# Triacylglycerols and Other Lipids Profiling of Hemp By-Products

**DOI:** 10.3390/molecules27072339

**Published:** 2022-04-05

**Authors:** Arjun H. Banskota, Alysson Jones, Joseph P. M. Hui, Roumiana Stefanova

**Affiliations:** Aquatic and Crop Resource Development Research Centre, National Research Council Canada, 1411 Oxford Street, Halifax, NS B3H 3Z1, Canada; alysson.jones@nrc-cnrc.gc.ca (A.J.); joseph.hui@nrc-cnrc.gc.ca (J.P.M.H.); roumiana.stefanova@nrc-cnrc.gc.ca (R.S.)

**Keywords:** hemp seed by-products, hemp seed oil, hemp cake, hemp hulls, triacylglycerols, fatty acids, DPPH radical scavenging activity

## Abstract

Hemp seed by-products, namely hemp cake (hemp meal) and hemp hulls were studied for their lipid content and composition. Total lipid content of hemp cake and hemp hulls was 13.1% and 17.5%, respectively. Oil extraction yields using hexane, on the other hand, were much lower in hemp cake (7.4%) and hemp hulls (12.1%). Oil derived from both hemp seeds and by-products were primarily composed of neutral lipids (>97.1%), mainly triacylglycerols (TAGs), determined by SPE and confirmed by NMR study. Linoleic acid was the major fatty acid present in oils derived from hemp by-products, covering almost 55%, followed by *α*-linolenic acid, covering around 18% of the total fatty acids. For the first time, 47 intact TAGs were identified in the hemp oils using UPLC-HRMS. Among them, TAGs with fatty acid acyl chain 18:3/18:2/18:2 and 18:3/18:2/18:1 were the major ones, followed by TAGs with fatty acid acyl chain of 18:3/18:3/18:2, 18:2/18:2/16:0, 18:2/18:2/18:1, 18:3/18:2.18:0, 18:2/18:2/18:0, 18:2/18:1/18:1 and 18:3/18:2:16:0. Besides TAGs, low levels of terpenes, carotenoids and cannabidiolic acid were also detected in the oils. Moreover, the oils extracted from hemp by-products possessed a dose-dependent DPPH radical scavenging property and their potencies were in a similar range compared to other vegetable oils.

## 1. Introduction

*Cannabis sativa* is an annual herbaceous plant that has been used as a source of food, fiber and medicine over centuries [1]. Three main cultivar groups of cannabis plants have been widely grown around the world for the production of industrial fibers, hemp seeds and cannabinoids, especially cannabidiol (CBD). Industrial hemp and cannabis (marijuana) are the two main *C. sativa* plants cultivated in Canada. According to Health Canada, 77,800 acres of industrial hemp were planted in 2018, mostly in the prairies (Alberta 38.5%, Saskatchewan 35% and Manitoba 155) [2]. Cultivation of hemp is expected to rise significantly in the coming years because of increasing demand for either hemp seed oil for food or hemp oil containing mainly CBD having potential medicinal value. It is forecasted that 450,000 acres of hemp will be planted in Canada in coming days with a worth of CAN 1 billion market value [2].

Production of a large quantity of hemp seed oil and hemp oil generates a huge amount of waste, including steam, leaves, roots and residual biomass obtained after oil extraction. Most of the hemp plant waste biomass ends up at landfills or as low-value products such as compost. Residual biomass collected after oil extraction from hemp seed commonly known as hemp cake or hemp meal is studied for its nutritional components and for possible bread supplementation because of its high protein content [3,4]. The overview of hemp by-products’ nutrients, phytochemical composition, bioavailability and bioefficacy was documented by Semwogerere et al., 2020 [5]. The hemp waste has also been examined for the production of biofuel or cement replacement in concrete production [6]. Since hemp seed oil is reported to have an excellent polyunsaturated fatty acids (PUFAs) profile, including both omega-3 (*ω*-3) fatty acids and omega-6 (*ω*-6) fatty acids [7,8], hemp cake oil also demonstrates a similar fatty acid profile possessing potential for aquafeed application [3].

Aquaculture is a well-established industry in Canada, with production activities occurring in every province and territory. To sustain current aquaculture farming, it is important to have new sources of feed ingredient inputs, especially lipid and protein. Fishmeals are primarily made from forage fish and by-products of the commercial fishery industry. With the global supply of forage fish at a plateau, the aquaculture industry has heavily shifted to use of plant-based proteins and oil to reduce dependency on conventional fishmeal [9,10]. The inclusion rate of dietary fishmeal and fish oil used within salmon feeds has significantly reduced to 15–18% and 12–13%, respectively, of the total diet [11]. In the present study we explore the lipid content of hemp waste, especially hemp cake and hemp seed hulls, and their characterization for their possible aquafeed application. Cold-pressed commercial hemp oil and oils extracted from hemp hearts and whole hemp seeds are also analyzed for comparison.

## 2. Results

### 2.1. Total Lipid Content and Oil Extraction Yield

Hemp by-products, primarily hemp cake and hemp hulls, together with hemp hearts and whole hemp seeds (Figure 1), were ground and flour with particle size <1.0 mm was collected using a laboratory sifter (Buhler AG, Uzwil, Switzerland). Total lipid content of the ground samples was measured by the Folch method [12] and results are shown in Table 1. Hemp hearts possessed the highest lipid content (54.7%), followed by the whole hemp seeds (48.0%). Even though hemp cake was obtained after oil extraction from hemp seeds, the biomass retains 13.1% total lipids. Hemp hulls also have 17.5% total lipids.

Hexane was used to extract oil from hemp by-products, hemp hearts and hemp seeds. Hemp hearts showed the highest oil extraction yield, 45.9%, followed by whole seed (36.4%). Both hemp hulls and hemp cake have a lower oil extraction yield of 12.1 and 7.4%, respectively. Oil extracted from hemp hearts has a light yellow color and oil from whole seeds has a light greenish color. Oils extracted from both hemp by-products and cold-pressed commercial oil are dark green in color (Figure 2).

### 2.2. Separation of Lipid Classes and ^1^H NMR Analysis

Oils derived from either hemp by-products or from hemp seeds together with cold-pressed hemp oil were further fractionated using solid phase extraction (SPE) to determine neutral lipid, glycolipids and phospholipids content gravimetrically [13]. All the oil samples contained more than 97.1% of neutral lipid, mostly triacylglycerols (TAGs). Percentage of glycolipids content ranged from 0.3 to 2.2%; similarly phospholipids were in the range of 0.3–1.3% (Table 1). The ^1^H NMR spectrum of the oils further shows an identical spectrum for all five oils derived either from hemp by-products or hemp seeds (Appendix A). The major signals observed in ^1^H NMR spectrum entirely belong to TAGs, either fatty acid acyl chain (0.75–3.00 and 5.70 ppm) or glyceride (4.0–4.25 and 5.23 ppm), strongly suggesting that hexane extract or cold-pressed commercial hemp oil was comprised mainly of TAGs [14].

### 2.3. Triacylglycerols (TAGs) Analysis of Neutral Lipid Using UPLC-HRMS

Ultra-high performance liquid chromatography high-resolution mass spectrometry (UPLC-HRMS) was used for the identification of individual TAGs present in the neutral lipid fraction of the oil extracted from hemp by-products and other hemp biomasses plus cold-pressed commercial hemp oil. The total ion current (TIC) chromatogram of all oil samples is shown in Figure 3. TAGs were eluted between 1.0 min and 5.00 min and almost identical TICs were observed for all the tested neutral lipid fractions derived from either hemp by-products or hemp seeds.

In total, 47 different TAGs were detected in oil either extracted from by-products or hemp seeds, and are listed in Table 2 with accurate mass measurement with retention time. Heat map analysis of individual TAG adduct ions [M + NH_4_]^+^ of individual neutral lipid fraction suggested that TAGs with fatty acid acyl chain 18:3/18:2/18:2 and 18:3/18:2/18:1 eluting around 1.95 min and 2.35 min, respectively, were the two major TAGs present in the oils extracted from all hemp seed biomasses. Concentration of TAGs with acyl side chain 18:3/18:3/18:2, 18:2/18:2/16:0, 18:2/18:2/18:1, 18:3/18:2.18:0, 18:2/18:2/18:0, 18:2/18:1/18:1 and 18:3/18:2:16:0 was also higher based on the relative intensities of the ammonium adduct ions. TAGs with long chain fatty acid side chain, including C20:0, C22:0 and C24:0 were relatively lower in concentration. Individual fatty acids of TAGs with acyl side chain length 59:8, 50:4, 59:6, 60:6 and 62:3 were unable to be identified because of the overlapping signal and the MS system only picked the three strongest peaks at the given retention time.

### 2.4. Fatty Acid Analysis

Fatty acid methyl esters (FAMEs) of the oil derived from both hemp by-products and other biomasses together with cold-pressed commercial oil were analyzed by gas chromatography (GC) after transesterification. The GC chromatogram is shown in Appendix A and the data are presented in Table 3. The total fatty acid in the oil ranged from 747.7 to 863.9 mg/g of the oil. Oil extracted from hemp cake had the highest fatty acid content, 863.9 mg/g, followed by cold-pressed hemp oil, 859.2 mg/g. Linoleic acid (C18:2 n-6) was the predominant fatty acid in all the analyzed oil which covers more than 54.9% of the total fatty acids. *α*-linolenic acid (C18:3 n-3) was the second dominant fatty acid followed by oleic acid (C18:1 n-9) and palmitic acid (C16:0). Steric acid, γ-linolenic acid and heptadecanoic acid are the other fatty acids detected in oil extracted from hemp by-products or other hemp seed samples. Small amounts of long chain fatty acids including arachidic acid (C20:0), *cis*-11-eicosenoic acid (C20:1), *cis*-11,14-eicosadienoic acid (C20:2), behenic acid (C22:0), erucic acid (C22:1) and lignoceric acid (C24:0) were also present, each covering less than 0.4% of total fatty acid.

### 2.5. Pigment Analysis

The oil extracted from hemp whole seeds, hearts, hulls, cake and cold-pressed hemp oil was analyzed for its pigment content, carotenoids and chlorophylls using high-performance liquid chromatography (HPLC). Lutein was detected in all tested hemp oil samples. The highest lutein content was observed in hemp cake, i.e., 0.125 mg/g oil. β-carotene and α-carotene were also detected in oil extracted from both hemp by-products and cold-pressed oil (Table 4). Several unidentified chlorophyll degradation peaks were detected in all analyzed oils except hemp hearts. HPLC chromatograms are shown in Appendix A.

### 2.6. Terpene and Cannabinoid Analysis

Gas chromatography mass spectrometry (GC-MS) was used to analyze terpenes present in oil extracted from hemp by-products and hemp seeds together with cold-pressed hemp oil. The TIC of the GC-MS is shown in Appendix A. A number of terpenes were detected in the oil sample derived either from hemp by-products or hemp seeds but their concentrations were below the limit of quantitation (LoQ). *α*-pinene, *β*-pinene, terpinolene, *β*-carophyllene and *α*-humulene were common terpenes detected in all tested oil. Delta-limonene, p-cymene, isopulegol and geraniol were detected only in oil derived from whole seeds (Table 4).

UPLC-HRMS was used for cannabinoid analysis in the oils extracted from hemp by-products and other hemp samples together with cold-pressed commercial hemp oil. Cannabinoid standards including cannabidiolic acid (CBDA) and CBD were used for calibration. A low level of CBDA was detected in oil derived from both hemp cake (0.027 mg/g) and hemp hulls (0.039 mg/g). CBD was detected in all tested oils except hemp hearts, and CBD concentration was below the limit of quantitation (LoQ). No other cannabinoids were detected in oils extracted from hemp by-products and hemp seed oils. The single ion monitoring of CBD and CBDA is shown in Appendix A for all hemp oils.

### 2.7. DPPH Radical Scavenging Activity

The percentage yields of the MeOH extracts of the oil derived from the hemp hearts, whole seeds, hemp cake, hump hulls and the cold-pressed commercial oil were 4.4, 6.6, 6.7, 11.0 and 4.4%, respectively. The MeOH extracts were tested for their DPPH radical scavenging activity at various concentrations. Hemp oils showed weak DPPH radical scavenging potency; IC_50_ values ranged from 555.0 to 3062.5 μg/mL as compared to ascorbic acid (IC_50_ −2.4 μg/mL), which was used as a positive control. All the tested oil showed dose-dependent DPPH radical scavenging activity (Figure 4). Hemp cake oil possessed the strongest DPPH radical scavenging activity with an IC_50_ value of 555.2 μg/mL followed by cold-pressed commercial hemp oil with an IC_50_ value of 610.0 μg/mL. The oil derived from the whole hemp seeds had the weakest DPPH radical scavenging activity with an IC_50_ value of 3062.5 μg/mL.

## 3. Discussion

Hemp seed oil is well-known for its health benefits because of its high polyunsaturated fatty acid (PUFA) content, especially linoleic acid (LA) *ω*-6 and α-linolenic (ALA) *ω*-3 acid [15,16,17]. The cold-pressed technique is commonly used to extract commercial hemp oil from seeds, leaving almost half of the total biomass as by-products commonly known as hemp cake or hemp meal. Hemp seed hulls are another hemp by-product obtained during the hulling process that produces hemp seed hearts, which can be consumed raw or cooked with other food. A number of studies have been conducted on hemp cake for its chemical composition and protein content [3,18,19], but no systematic study has been conducted on hemp seed hulls for their chemical characterization. The current study focuses on in-depth lipid analysis of hemp by-products, especially hemp cakes and hemp hulls, for their possible aquafeed application. Total lipid content of both hemp cake and hemp hulls was determined together with the oil extraction yield using hexane as extracting solvent. Whole hemp seeds and hemp hearts together with cold-pressed commercial hemp oils were also studied simultaneously for comparison.

The lipid content of whole hemp seeds (48.0%) was slightly lower than hemp hearts (54.7%) because of the hull’s presence, which has 17.5% total lipid. Even though the hemp cake was collected after cold-pressed oil extraction, it still contained 13.1% total lipid, similar to the previous study by Pojic et al., 2014 [3]. The hemp meal fractions reported to have oil content varied from 8.26 to 18.6% depending on the partial size. In the current study, we have used hexane as extracting solvent because it has less toxicity, is easy to evaporate and is widely used in food industries for oil extraction. The results suggested that hemp cake has the lowest hexane extractable oil, yielding only 7.4% of the biomass, which is almost half of the total lipid content (13.1%). The oil extraction yield of hemp seed hulls, hemp hearts and hemp whole seeds was also lower than the total lipid content (Table 1). Lower oil extraction yield suggested there should be polar lipids such as glycolipids or phospholipids in the residual biomass after hexane extraction; further study is needed to characterize such lipids in hemp seeds and in hemp seed by-products. Solid phase extraction (SPE) technique with silica gel cartridge was further used to determine the percentage of various classes of lipids present within the oil extracted by hexane [13]; the SPE results suggested that oils (hexane extract) extracted from hemp by-products are primarily composed of neutral lipids, which comprise >97.1% of the oil (Table 1). The proton NMR signals in their respective spectra displayed peaks only belonging to TAGs (Appendix A), i.e., either fatty acid acyl side chain or glyceride moiety [14]. The NMR spectral signals clearly demonstrated that the oil extracted either from by-products or hemp seeds is made up of mainly triglycerides (TAGs).

Hemp seed and hemp seed meal have been well-studied for their fatty acid composition; results suggested that LA, ALA, oleic acid, γ-linolenic acid and palmitic acid are the main fatty acids present in hemp seed or hemp meal oil [7,8,17,20]. LA (C18:2 n-6) is the major fatty acid that covers more than 50% of the total fatty acids and its percentage varies depending upon the extraction methods and cultivars. Oomah et al. (2002) reported 53.4–56.6% of LA in hemp seed oils extracted with petroleum ether using the Soxhlet extraction method [21]. Similarly, 58% LA was reported in hemp seed oil extracted with supercritical carbon dioxide (CO_2_) [20] and 55.2–54.8% LA was reported in hemp seed meal fractions [3]. In the current study, LA content in the hemp by-products oils was 54.9–55.9% of the total fatty acids. α-linolenic acid (ALA) was the second major fatty acid in the oil extracted from hemp by-products, covering 16.7–18.4% of the total fatty acids. Longer chain fatty acids, i.e., C:20 or higher, were also detected but their concentration was significantly lower than LA and ALA. Moreover, PUFA *ω*-6 covers almost 60% and PUFA ω-3 covers only 18% of the total fatty acid count in the oil derived from hemp by-products. Almost identical results were observed for the oil derived from hemp hearts, hemp seeds and cold-pressed commercial hemp oil.

Dietary lipids play important roles as a source of energy and essential fatty acids necessary for fish growth and development [22]. There is urgent need for alternative sources of protein and lipid diet in the aquaculture industry because of increasing demand for aquaculture products, especially fish [9,10]. In general, freshwater fish require either LA (18:2 *ω*-6), ALA (18:3 *ω*-3) or both, whereas marine fish require long chain fatty acids, eicosapentaenoic acid (EPA, 20:5 *ω*-3) and/or docosahexaenoic acid (DHA, 22:6 *ω*-3) [23]. Oils extracted from hemp by-products, namely hemp cake and hulls, contain high levels of LA and ALA, obviously suitable for freshwater aquafeed application. Extensive research has been conducted to study the effects of alternative lipid sources, mainly vegetable oils, on the growth of both marine and freshwater aquatic animals [24,25]. Even though no growth performance advantage was observed in juvenile *Tor tambroides* with LA/ALA treatment against palm oil which is mainly composed of saturated fatty acids [26], significantly higher concentrations of *ω*-3 PUFA were found in the muscle of the LA/ALA feed diet group [27]. Studies on Atlantic salmon suggested that partial replacement of fish oil by vegetable oils (40–60%) can produce similar results as diets containing 100% fish oil during the grow-out phase of Atlantic salmon in the sea and does not significantly affect *ω*-3 fatty acid composition in muscle when feeding fish with a diet containing low levels of fish meal and moderate levels of fish oil [28,29], suggesting the use of vegetable oil having little or no EPA/DHA for marine aquafeed application. In another study, Xue et al. (2006) observed that the fatty acid composition of fish fillets and livers reflected the dietary fatty acid composition when studying six alternative lipid sources in Japanese sea bass (*Lateolabrax japonicas*) [30]. Sankian et al. (2019) demonstrated that the complete replacement of fish oil is even possible in mandarin fish diets [31]. These studies clearly demonstrate vegetable oils are already starting to replace fish oil either partially or fully in fish feed diets. Soybean is one of the well-studied vegetable oils for aquafeed, having high levels of PUFA LA and ALA, containing 55% and 13%, respectively [32]. Oil derived from hemp by-products has a slightly better *ω*-3 profile than soybean oil because it has higher ALA (18%). Possible use of hemp seed by-products oil for aquafeed application not only helps in managing the waste biomass, it also gives economic value for emerging hemp industries around the world. In addition, stearidonic acid (SDA), which is also present in hemp by-products, is an intermediate fatty acid in the biosynthetic pathway from α-linolenic acid to EPA/DHA, and the conversion from SDA is more efficient than from α-linolenic acid [33]. The ω-6/ω-3 fatty acid ratio is another factor which plays an important role in health benefits for both humans and aquatic animals. Human beings evolved on a diet with a ratio of ω-6/ω-3 essential fatty acids of 1 whereas in Western diets the ratio is 15/1 to 16.7/1; a lower ratio of ω-6/ω-3 is needed for the prevention and management of chronic diseases [34]. A study suggested that an increase in the dietary ω-6/ω-3 ratio was reported to influence the growth of Atlantic salmon challenged with *Paramoeba perurans* [35]. The ω-6/ω-3 fatty acid ratio of oils extracted from hemp cake and hemp hulls was 3.0:1 and 3.1:1, respectively. Several factors such as climatic conditions, cultivars, nutrients, etc. could affect the ω-6/ω-3 fatty acid ratios of hemp oil and their ratios were reported as 1.71:1–3.31:1 [36].

Even though fatty acids are a key component of the hemp seed oils and oils derived from hemp seed by-products, only a small amount may exist in free acid forms. The majority of fatty acids are present in ester forms, either in triacylglycerols (TAGs) or in polar lipids such as glycolipids or phospholipids. It is worth mentioning here that bioavailability of the fatty acids depends upon the structural composition of triacylglycerols. It has been reported that re-esterified triacylglycerols have superior bioavailability of EPA and DHA as compared to the natural fish oils, ethyl esters or free fatty acid forms [37]. It is crucial to know the structure composition of the intact lipids (or triacylglycerols) to understand their true health benefits for humans or as feedstock, including aquafeed application. To the best of our knowledge, no research has been conducted to date on characterization of intact TAGs in hemp seed oil or oil-derived hemp by-products. In the current study, electrospray ionization (ESI) was used to characterize TAGs in neutral lipid fractions of oil derived from hemp by-products. ESI is a soft ionization technique previously used by MacDougall at al. (2011) for profiling TAGs of microalgae [38]. ESI generates intact molecular ions, typically observed as ammonium, sodium or proton adducts (Figure 5). The ammonium adducts ion was selected for identification of TAGs using the LipidMAPS database [39]. Moreover, the molecular ion of TAG could be fragmented under low-energy collision-induced dissociation (CID) in a systematic fashion into diacylglycerol (DAG) product ions which were used for the identification of TAGs. Representative mass spectra of four TAGs having fatty acid side chains 18:3/18:2/18:1, 18:2/18:1/16:0, 24:0/18:3/18:2 and 22:0/18:2/18:1 with their fragment ions are shown in Figure 5. Based on the accurate masses of adduct and fragment ions, we have for the first time identified 47 TAGs in the oil derived from hemp by-products or hemp seeds. The MS spectrum in Appendix A showed that the precursor ion at *m*/*z* 896.77087, 870.75549 and 844.74030 eluted at 2.35 min of hemp cake belongs to TAGs. Even though some of these TAGs were not chromatographically resolved, their identifications were incontrovertible. The major fragments observed in the MSMS spectra represent the neutral losses of fatty acids from the glyceride backbone. For example, in Figure 5, the precursor ion at *m*/*z* 896.77156 first lost an ammonium adduct and further lost major fragments at *m*/*z* 601.51970, *m*/*z* 599.50386 and *m*/*z* 597.48850, representing the neutral loss of fatty acids C18:3, C18:2 and C18:1, respectively. Unfortunately, determination of the position of individual fatty acyl chains within glyceride backbone was outside the scope of this study. Based on the intensities of the ammonium adducts ions we were able to create a hit map of the individual TAGs within the neutral lipid fractions to compare their relative concentration. The hit map diagram of individual TAGs (Table 2) strongly suggested that TAGs with fatty acyl chain 18:3/18:2/18:2 and 18:3/18:2/18:1 are major components in all the tested hemp oils derived either from hemp by-products or from hemp seeds. Fatty acid analysis data (Table 3) clearly demonstrated that linoleic acid (C18:2 n-6), *α*-linolenic acid (C18:3 n-3), oleic acid (C18:1 n-9), stearic acid (C18:0) and palmitic acid (C16:0) were the major fatty acids in hemp oil. This further confirmed that the TAGs having these fatty acyl chains (C18:3, C18:2, C18:1, C18:0 and C16:0) are the main TAGs in the hemp oils. TAGs with longer fatty acid acyl chains, i.e., C:20, C:22, C:23 and C:24 were also detected, but their intensities were lower as compared to the TAGs containing major fatty acid acyl chains.

Carotenoids, chlorophyll and cannabinoids, especially CBD, were also reported in hemp seed oil [36]. In the current study, we also found the presence of lutein in all oil samples extracted from either by-products or hemp seeds and cold-pressed commercial hemp oil. The light yellow color of the hemp hearts oil may be due to low levels of lutein. *α*-carotene and *β*-carotene were also detected in oils derived from hemp cake and hemp hulls plus cold-pressed hemp oil. Chlorophyll degradation products and cannabinoids, especially CBD and CBDA were observed in most of the oils except hemp hearts and whole seeds. Absence of cannabinoids and chlorophyll and only low levels of lutein presence in the hemp hearts oil clearly suggested that the cannabinoids, carotenoids and chlorophyll present only in the hemp hulls. Higher concentrations of pigments and chlorophyll in hemp cake indicated that the cold-pressed process enhances the extraction efficiency of pigments. Lower levels of lutein and absence of other carotenoids in whole seeds oil suggested that the heating process may possibly degrade the pigments. Terpenes were also present in all hemp samples, with more terpenes being detected in whole seed, cake and cold-pressed hemp oil. Hemp hearts and hemp hulls had fewer terpenes detected. Hemp seeds have been reported to have antinutrient components in them such as phytic acid or phytate [40,41]. Even though we did not analyze phytic acid or phytate in oils extracted from hemp seeds and by-products, these antinutrient components are most likely not present in the hemp seed oils based on their polarity and ability to be extracted by hexane.

Vegetables oil has been reported to have antioxidant properties [42]; thus, we further tested oils derived from both hemp cake and hemp hulls for their DPPH radical scavenging potency for comparison with other vegetable oils. Oils extracted from other hemp biomass and cold-pressed hemp oils were also tested. Cold-pressed hemp oil and oil derived from hemp cake possessed the highest DPPH radical scavenging potency compared to the oil extracted from hemp hearts and hemp hulls, which had relatively higher concentrations of α-carotene and β-carotene. The oil derived from hemp seeds showed significantly weaker DPPH radical activity as compared to other oils; most probably, the roasting process destroyed the antioxidative component. The mechanical crushing process seems to enhance extraction of antioxidative components present in the hemp oils. The IC_50_ values of the tested oils range from 555 to 3062.5 μg/mL, weaker as compared to the positive control ascorbic acid having IC_50_ 2.4 μg/mL. The IC_50_ values of hemp cake and cold-pressed hemp oils are in a similar range as those of soybean oil (IC_50_ 362 μg/mL), canola oil (IC_50_ 400 μg/mL), sesame oil (IC_50_ 500 μg/mL) and are better than flax seed oil (IC_50_ 2400 μg/mL) and avocado oil (IC_50_ 6200 μg/mL) [42].

In summary, we have studied lipid composition of oils derived from hemp seed hulls and hemp cake together with whole hemp seeds, hemp hearts and commercial cold-pressed hemp oil. Both hemp by-products, i.e., hemp hulls and hemp cakes contain significant amounts of total lipids, 17.5 and >13.1% of the biomass, respectively, having great commercial value as a source of lipid for aquafeed application. It is worth mentioning here that this is the first report of such comprehensive lipid characterization for oil derived from hemp seed hulls and hemp cake in comparison with cold-pressed commercial oil and oil extracted from hemp whole seeds and hemp hearts. The fatty acids profile suggested that, regardless of material, for either hemp seed or hemp by-products, all biomasses had almost identical fatty acid compositions. LA and ALA acids were the two dominant fatty acids present in the oil extracted from hemp by-products, similar to those of cold-pressed hemp oil. The total *ω*-6/*ω*-3 ratios of the tested hemp oils were 3.1 to 3.3. For the first time, we have characterized the triacylglycerol content of the hemp oil and 47 individual TAGs were identified by UPLC-HRMS analysis. Among them, TAGs with fatty acyl side chain 18:3/18:2/18:2 and 18:3/18:2/18:1 were the major ones. A low level of carotenoids, especially lutein, *α*-carotene and *β*-carotene was found in oils extracted from hemp by-products. Moreover, the oils derived from hemp by-products possessed dose-dependent DPPH radical scavenging properties and their potency was in a similar range as compared to other vegetable oils such as canola, soybean, avocado and sesame oils.

## 4. Materials and Methods

### 4.1. General

The NMR spectra were measured on a Bruker 500 or 700 MHz spectrometer with deuterated chloroform. HPLC analysis was carried out on an Agilent 1200 Series HPLC equipped with a diode array detector. High-resolution mass spectra were recorded with a Thermo Fisher Scientific (San Jose, CA, USA) Q Exactive mass spectrometer. HPLC grade solvents and Milli-Q water were used for the extraction, fractionation and LC/MS analysis.

### 4.2. Research Materials

Hemp cake, hemp seed hulls, hemp hearts and cold-pressed hemp oil were received from Fresh Hemp Food Ltd., Winnipeg, Manitoba, Canada. Roasted whole hemp seeds were purchased from a local store (Bulk Barn) in Halifax, Nova Scotia, Canada. Samples were stored at room temperature after being received, except the hemp oil, which was stored in a freezer (−20 °C) prior to extraction or LC/MS analysis. All research materials were pulverized using a mechanical grinder and were filtered through a laboratory sifter (Buhler AG, Uzwil, Switzerland) to collect flour with particle size <1.0 mm for further analysis.

### 4.3. Total Lipid Content and Oil Extraction

Total lipid content was determined by the Folch method with slight modifications [11]. In brief, a pulverized sample (~100 mg) was extracted at room temperature, homogenizing with CHCl_3_/MeOH (2:1, 1 mL × 3) using a bead beater (Bead Mill_24_, Fisher Scientific, Hampton, NH, USA) in 2 mL Lysing matrix Y tubes (3 × 1 min cycles) in triplicate. The combined lipid extracts were dried under nitrogen and kept under vacuum overnight; weight was measured gravimetrically and total lipid content was calculated using the following formula.
Total lipid (%) = weight of lipid/weight of sample × 100

The pulverized sample was further extracted with hexane to collect oil from both hemp sample and hemp wastes. The solid/solvent ratio was 1:20 (*w*/*v*) and the 20 g sample was used for oil extraction. Extraction was performed at room temperature overnight.

### 4.4. Lipid Class Separation by Solid Phase Extraction (SPE)

The oils (hexane extract) either extracted from hemp sample or received from Hemp Oil Canada were further fractionated into three different classes of lipid, i.e., neutral lipids containing mostly triacylglycerols (TAGs), glycolipids and phospholipids following the previously reported solid phase extraction (SPE) method by Ryckebosch et al., 2011 [13]. Briefly, the SPE column (Discovery DSC-Si Tube 3 mL 500 mg) was conditioned with 10 mL chloroform. Approximately 100 mg oil in 1.0 mL chloroform was applied to the column. The column was then eluted successfully with chloroform (10 mL), acetone (10 mL) and methanol (10 mL), yielding neutral lipid, glycolipid and phospholipid, respectively. The percentage of each class of lipid was determined by their weight taken gravimetrically after drying under nitrogen followed by being under vacuum overnight.

### 4.5. Triacylglycerols (TAGs) Analysis by UPLC-HRMS

UPLC-HRMS data were acquired on an UltiMate 3000 system coupled to a Q-Exactive hybrid Quadrupole Orbitrap Mass Spectrometer (Thermo Fisher Scientific, Waltham, MA, USA) equipped with a HESI-II probe for electrospray ionization. Separation was achieved on a Thermo Hypersil Gold C8 column (100 × 2.1 mm, 1.9 μm) at 40 °C. Through a flow-splitter, approximately 1/15 of LC eluent was sent to the mass spectrometer. A makeup solution consisting of 5 mM ammonium formate in IPA/de-ionized/methanol 1/2/7 (*v*/*v*) was delivered constantly at 100 µL/min to MS. The solvent system was composed of acetonitrile and IPA. Initial gradient was 100% acetonitrile, which increased linearly to 5% IPA in 1 min, and then linearly to 70% IPA in 8 min, and was held for 2 min, at a flow-rate of 750 µL/min.

MS data were acquired in positive ion mode in data dependent mode, alternating between full MS and MSMS scans, where the three most abundant precursor ions were subjected to MSMS using 25 eV collision energy. The source parameters were set as follows: sheath gas: 15, auxiliary gas flow: 4, sweep gas: 0, spray voltage: 2.1 kV, capillary temperature: 375 °C, heater temperature: 300 °C.

### 4.6. Fatty Acid Analysis by GC

Fatty acid analysis was performed according to the AOAC official method 991.39 (AOAC, 2000) with slight modifications in triplicate [43]. Briefly, ~10 mg of oil extracted from hemp biomasses or cold-pressed commercial hemp oil was placed in a dry 5 mL screw-capped reaction vial with MeOH (1.0 mL) containing 0.1 mg methyl tricosanoate as an internal standard (IS). The mixture was sonicated and 1.5 N NaOH solution in MeOH (0.5 mL) was added; blanketed with nitrogen; heated for 5 min at 100 °C and cooled for 5 min. BF_3_ 14% solution in MeOH (1.0 mL, Sigma-Aldrich, St. Louis, MI, USA) was added, mixed, blanketed with nitrogen and heated at 100 °C for 30 min. After cooling, the reaction was quenched by the addition of water (0.5 mL) and the FAME was extracted with hexane (2.0 mL). Part of the hexane layer (300–600 μL) was transferred to a GC vial for analysis by GC-FID. GC-FID was carried out on an Agilent Technologies 7890A GC spectrometer using an Omegawax 250 fused silica capillary column (30 m × 0.25 mm × 0.25 μm film thicknesses) for fatty acid analysis. Supelco^®^ 37 component FAME mix and PUFA-3 (Supelco, Bellefonte, PA, USA) were used as fatty acid methyl ester standards. Fatty acid content in hemp oil samples was calculated by the following equation and expressed as mg/g sample.
Fatty acid (mg/g) = (A_X_ × W_IS_ × CF_x_/A_IS_ × W_S_ × 1.04) × 1000
where A_X_ = area counts of fatty acid methyl ester; A_IS_ = area counts of internal standard (tricosylic acid methyl ester); CF_X_ = theoretical detector correlation factor is 1; W_IS_ = weight of IS added to sample in mg; W_S_ = sample mass in mg; and 1.04 is factor necessary to express result as mg fatty acid/g sample.

### 4.7. Pigment Analysis

The oils (hexane extract) were dissolved in MeOH as 1 mg/mL concentration by sonicating for 10 min and were filtered through. The MeOH soluble fraction after filtration was used for HPLC analysis. Pigment including carotenoids and chlorophyll analysis was performed using an Agilent 1200 series HPLC with a YMC Carotenoid column (5 μm, 2 × 250 mm, 181 YMC Co. Ltd., Tokyo, Japan) eluting with 50 mM NH_4_OAc in MeOH/tertiary butyl methyl ether (TBME) linear gradient 5 to 65%B in 30 min at 0.2 mL min^−1^ flow rate for 60 min. Standard curves for chlorophyll a, chlorophyll b, astaxanthin, α-carotene, β-carotene, canthaxanthin, fucoxanthin, lutein, lycopene and zeaxanthin at 450 nm were used for carotenoids quantification.

### 4.8. Terpene and Cannabinoids Analysis

Oil extracted from hemp biomasses or cold-pressed hemp oil (1.0 g) was dissolved in 5 mL of ethyl acetate. Samples were diluted by half and spiked with 20 μg/mL of dodecane internal standard for GC-MS analysis. GC-MS analysis was carried out on an Agilent Technologies 6890N GC equipped with an Agilent 5975 MSD and FID detector. Separation was carried out using a Restek Rxi 625 Sil MS column (30 m × 2.5 mm × 1.4 µM film thickness). Restek^®^ Cannabis Terpenes Standard #1 containing 19 terpenes was used as a terpene standard. Samples were injected in triplicate. Terpene standard curve containing 1, 2.5, 5, 10 and 25 μg/mL of terpene standard and 20 μg/mL of dodecane internal standard was injected in duplicate for quantification and identification.

For cannabinoids analysis, hemp oils were dissolved in MeOH 1.0 mg/mL concentration and sonicated for 5 min at room temperature. The MeOH soluble part was applied to a C-18 solid phase extraction cartridge to remove residual triacylglycerols. The eluent was collected and further diluted ×10 and subjected to UPLC/HRMS, using Acquity UPLC HSS-T3 column (Waters, 2.1 × 100 mm 1.8 μm) with flow rate of 0.4 mL/min (0.1% formic acid in water/0.1 formic acid in MeOH with linear gradient; 20:80–16:84 in 2 min, 16:84–14:86 in 4 min) in full MS scan. The source parameters were as follows: sheath gas: 50, auxiliary gas flow: 10, spray voltage: 3.0 kV, capillary temperature: 300 °C, heater temperature: 300 °C. Standard cannabinoids purchased from Sigma-Aldrich, St. Louis, MI, USA were used as control.

### 4.9. DPPH Radical Scavenging Activity

1,1-Diphenyl-picrygydrazyl (DPPH) radical scavenging activity was performed according to the procedure described by Hatano et al. (1989) with minor modifications [44]. Oil derived from both hemp seeds and by-products (25 mL ≅ 22.7 g) was extracted with MeOH (150 mL × 2) by stirring at room temperature for 3 h. The MeOH soluble part was evaporated to dryness and used for DPPH radical scavenging assay. In brief, 100 µL of extract at various concentrations was mixed with an equal volume of 60 µM DPPH solution in MeOH, the resulting solution was thoroughly mixed and absorbance was measured at 520 nm after 30 min using a Spectra_max_ Plus Spectrophotometer plate reader (Molecular Devices, San Jose, CA, USA). The scavenging activity was determined by comparing the absorbance with that of controls containing only DPPH and MeOH (100%). Vitamin C, a known antioxidant, was used as a positive control. Measurements were carried out in triplicates.
DPPH Radical Scavenging activity (%) = 100 − (A_SD_ − A_SM_)/(A_DM_ − A_ME_) × 100
where A_SD_ = absorbance of sample + DPPH; A_SM_ = absorbance of sample + MeOH; A_DM_ = absorbance of DPPH + MeOH; and A_ME_ = absorbance of MeOH at 520 nm. IC_50_ values were also calculated from the linear range of each assay.

## Figures and Tables

**Figure 1 molecules-27-02339-f001:**
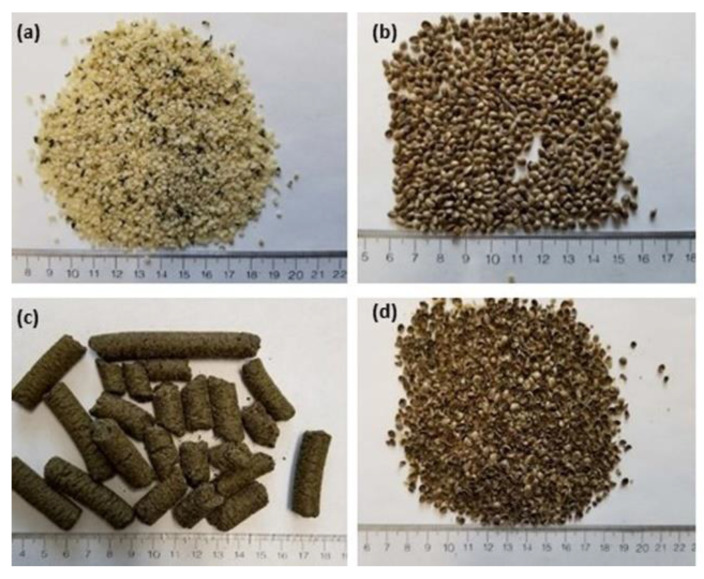
Research material. (**a**) hemp hearts, (**b**) whole hemp seeds, (**c**) hemp cakes and (**d**) hemp seed hulls.

**Figure 2 molecules-27-02339-f002:**
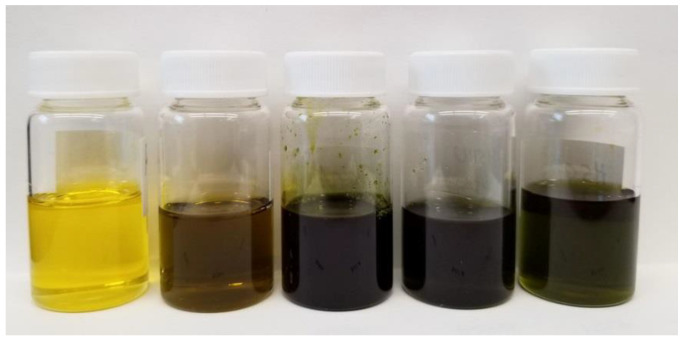
Hexane extract (oil), left to right—hemp hearts, whole seeds, hemp cake, hemp hulls and cold-pressed oil.

**Figure 3 molecules-27-02339-f003:**
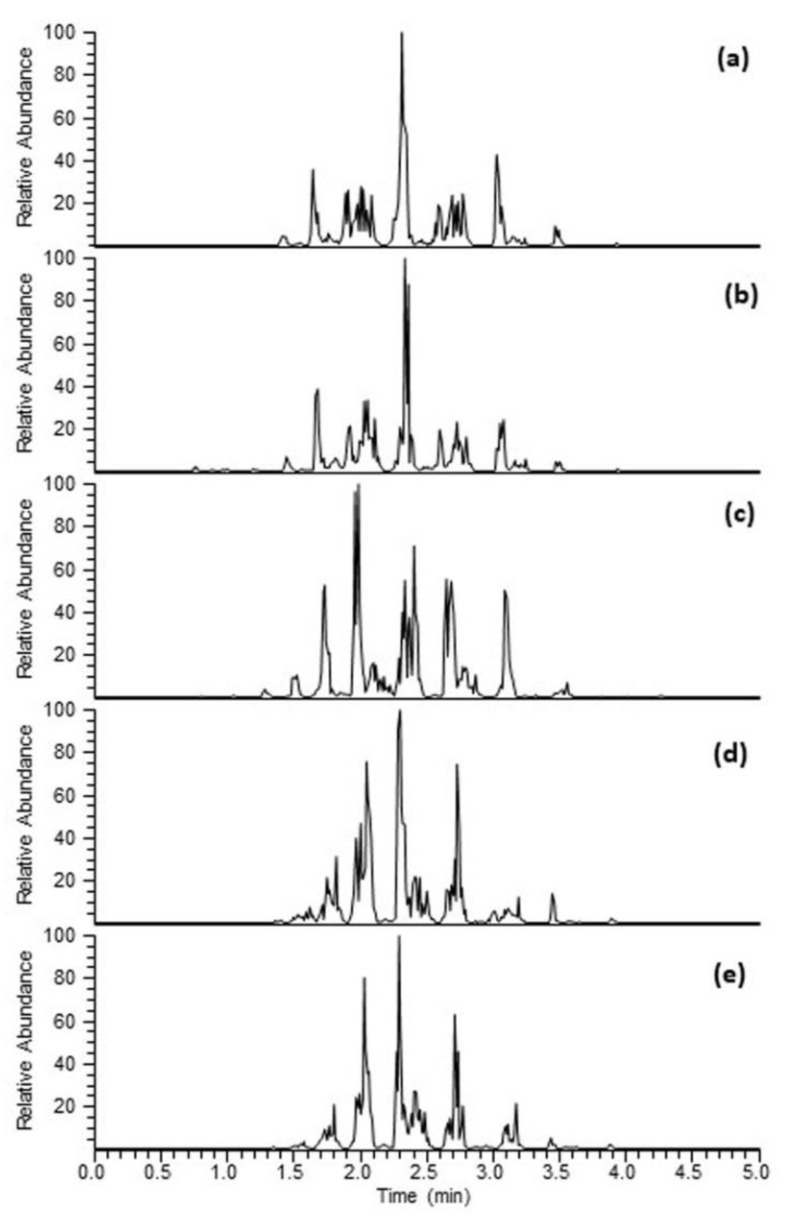
Base peak chromatograms of neutral lipid fraction of oil derived from hemp seeds, hearts, hulls, cake and hemp oil in positive mode ESI-MS. (**a**) Hemp hearts, (**b**) hemp cake, (**c**) hemp seed hulls, (**d**) hemp oil and (**e**) hemp whole seeds.

**Figure 4 molecules-27-02339-f004:**
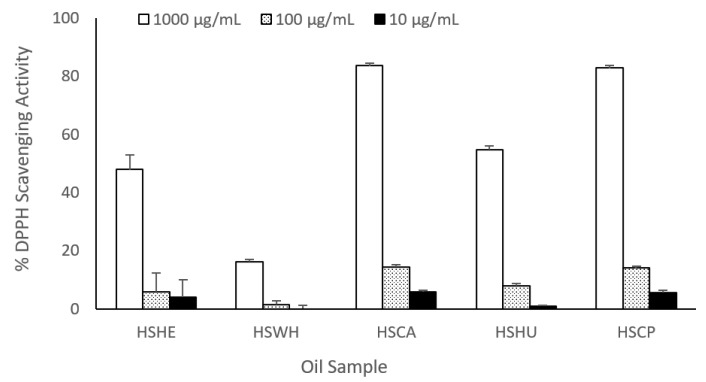
DPPH radical scavenging capacities of oil extracted from hemp hearts (HSHE), hemp whole seeds (HSWH), hemp cake (HSCA), hemp seed hulls (HSHU) and cold-pressed hemp oil (HSCP).

**Figure 5 molecules-27-02339-f005:**
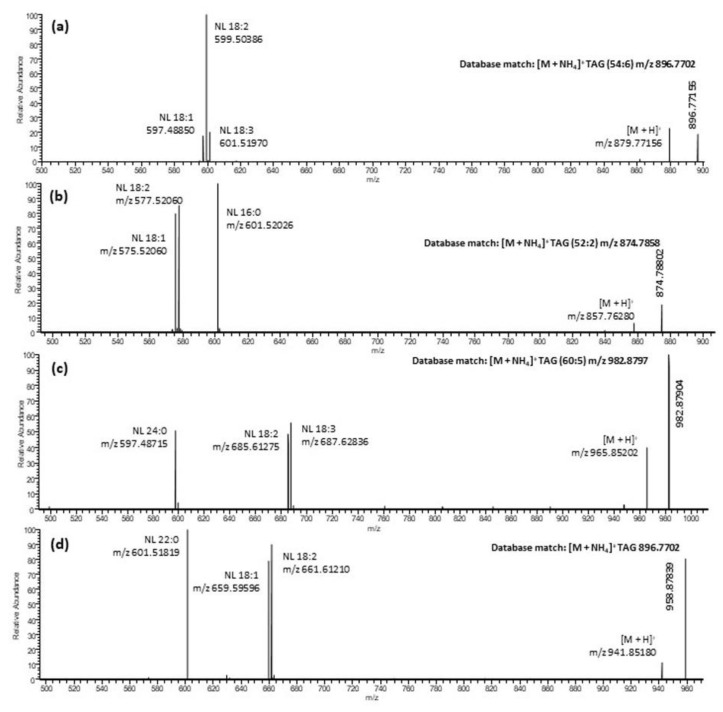
Representative mass spectrum of TAGs with fragmentation ions corresponding to neutral loss (NL). (**a**) TAG (54:6) 18:3/18:2/18:1, (**b**) TAG (52:3) 18:2/18:1/16:0, (**c**) TAG (60:5) 24:0/18:3/18:2 and (**d**) TAG (58:3) 22:0/18:2/18:1 with their fragmentation ions.

**Table 1 molecules-27-02339-t001:** Total lipid and oil extraction yield of hemp hearts (HSHE), hemp whole seeds (HSWH), hemp cake (HSCA) and hemp seed hulls (HSHU). Separation of lipid classes using solid phase extraction (SPE) expressed in percentage of oil.

Sample/Lipid Content	HSHE	HSWH	HSCA	HSHU	HSCP ^1^
Total lipid (%) ^2^	54.7 ± 2.3	48.0 ± 2.8	13.1 ± 0.1	17.5 ± 0.1	n/a
Oil (%)	45.9	36.4	7.4	12.1	n/a
Neutral lipids (%)	99.4	97.3	97.8	97.1	98.9
Glycolipids (%)	0.3	1.4	1.5	2.2	0.7
Phospholipids (%)	0.3	1.3	0.7	0.7	0.4

^1^ Cold-pressed hemp oil. ^2^ Results are from triplicate experiments.

**Table 2 molecules-27-02339-t002:** Heat map of triacylglycerols (TAGs) identified in oil extracted from hemp hearts (HSHE), hemp whole seeds (HSWH), hemp cake (HSCA), hemp seed hulls (HSHU) and cold-pressed hemp oil (HSCP).

HSHE	HSWH	HSCA	HSHU	HSCP	RT (min)	Measured (*m*/*z*)	Calculated (*m*/*z*)	Error (ppm)	C:DB	TAG Identity
					1.35	888.70819	888.70756	0.71	54:10	18:4/18:3/18:3
					1.45	890.72369	890.72321	0.54	54:9	18:4/18:3/18:2
					1.70	962.81750	962.81711	0.41	59:8	NI
					1.70	892.74231	892.73886	3.86	54:8	18:3/18:3/18:2
					1.70	866.72357	866.72321	0.42	52:7	18:4/18:3/16:0
					1.95	894.75446	894.75451	0.06	54:7	18:3/18:2/18:2
					1.95	868.74005	868.73886	1.37	52:6	18:3/18:3/16:0
					1.95	922.78534	922.78581	0.51	56:7	20:0/18:4/18:0
					2.35	896.76996	896.77016	0.22	54:6	18:3/18:2/18:1
					2.35	870.75519	870.75451	0.78	52:5	18:3/18:2/16:0
					2.35	844.73926	844.73886	0.47	50:4	NI
					2.35	966.84924	966.84841	0.86	59:6	NI
					2.56	872.76996	872.77016	0.23	52:4	18:2/18:2/16:0
					2.70	898.78320	898.78581	2.90	54:5	18:2/18:2/18:1; 18:3/18:2/18:0
					2.70	924.80255	924.80146	1.18	56:6	20:1/18:3/18:2
					2.91	886.78650	886.78581	0.78	53:4	18:2/18:2/17:0
					3.15	900.80005	900.80146	1.57	54:4	18:2/18:2/18:0; 18:2/18:1/18:1
					3.15	874.78625	874.78581	0.50	52:3	18:2/18:1/16:0
					3.15	848.77167	848.77016	1.78	50:2	18:2/16:0/16:0
					3.15	926.81543	926.81711	1.81	56:5	20:1/18:2/18:2; 20:0/18:3/18:2
					3.40	928.83148	928.83276	1.38	56:4	20:0/18:2/18:2
					3.40	876.80115	876.80146	0.35	52:2	18:2/18:0/16:0; 18:1/18:1/16:0
					3.40	902.81738	902.81711	0.30	54:3	18:2/18:1/18:0
					3.40	954.84802	954.84841	0.41	58:5	22:0/18:3/18:2
					3.70	980.86426	980.86406	0.20	60:6	NI
					3.70	968.86395	968.86406	0.11	59:5	23:0/18:3/18:2
					3.70	904.83282	904.83276	0.07	54:2	20:0/18:2/16:0
					3.85	930.84845	930.84841	0.04	56:3	20:0/18:2/18:1
					3.95	956.86414	956.86406	0.08	58:4	22:0/18:2/18:2
					3.95	982.87933	982.87971	0.39	60:5	24:0/18:3/18:2
					4.17	970.88013	970.87971	0.43	59:4	23:0/18:2/18:2
					4.21	932.86420	932.86406	0.15	56:2	22:0/18:2/16:0; 20:0/18:1/18:1; 20:0/18:2/18:0
					4.21	958.87958	958.87971	0.14	58:3	22:0/18:2/18:1
					4.21	984.89496	984.89536	0.41	60:4	24:0/18:2/18:2
					4.35	958.87994	958.87971	0.24	58:3	20:3/20:0/18:0
					4.35	932.86432	932.86406	0.28	56:2	20:0/18:1/18:1
					4.84	960.89569	960.89536	0.34	58:2	22:0/18:1/18:1
					4.84	986.91125	986.911	0.25	60:3	24:0/18:2/18:1
					5.00	988.92682	988.92666	0.16	60:2	24:0/18:1/18:1
					5.00	896.77002	896.77016	0.16	54:6	18:3/18:2/18:1
					5.00	1014.94234	1014.94231	0.03	62:3	NI

NI—Fatty acid was not identified. Low Abundance 
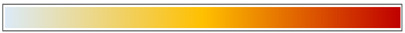
 High Abundance.

**Table 3 molecules-27-02339-t003:** Fatty acid profile of oil extracted from hemp hearts (HSHE), hemp whole seeds (HSWH), hemp cake (HSCA), hemp seed hulls (HSHU) and cold-pressed hemp oil (HSCP). Results are expressed in mg/g biomass and the percentage of individual fatty acid in the oil is given in the parenthesis.

Fatty Acid (FA)	HSHE	HSWH	HSCA	HSHU	HSCP
**Myristic acid (C14:0)**	-	-	-	-	-
**Myristoleic acid (C14:1)**	-	-	-	-	-
**Pentadecanoic acid (C15:0)**	-	-	-	-	-
***cis*-10-Pentadecenoic acid (C15:1)**	-	-	-	-	-
**Palmitic acid (C16:0)**	47.2 ± 12.0 (5.8)	45.6 ± 4.3 (6.1)	55.2 ± 2.9 (6.4)	55.1 ± 0.6 (6.8)	49.0 ± 0.4 (5.7)
**Palmitoleic acid (C16:1 n-7)**	-	-	-	-	-
**C16:2 n-4**	-	-	-	-	-
**C17:0 (Heptadecanoic acid)**	0.3 ± 0.2 (0.0)	0.1 ± 0.2 (0.0)	-	0.6 ± 0.0 (0.1)	-
**C16:3 n-4**	-	-	-	-	-
***cis*-10-Heptadecenoic acid (C17:1)**	-	-	-	-	-
**C16:4 n-1**	-	-	-	-	-
**Stearic acid (C18:0)**	21.1 ± 0.5 (2.6)	18.4 ± 1.8 (2.5)	20.7 ± 1.1 (2.4)	18.8 ± 0.2 (2.3)	20.8 ± 0.2 (2.4)
**Oleic acid (C18:1 n-9)**	33.0 ± 57.2 (11.9)	80.3 ± 7.8 (10.7)	86.4 ± 5.1 (10.0)	79.3 ± 0.8 (9.7)	81.2 ± 0.6 (9.5)
***cis*-Vaccenic acid** **(C18:1 n-7)**	5.7 ± 0.2 (0.7)	6.3 ± 0.6 (0.8)	7.9 ± 0.4 (0.9)	7.5 ± 0.1 (0.9)	6.8 ± 0.1 (0.8)
**Linoleic acid (C18:2 n-6)**	455.9 ± 10.9 (56.9)	417.8 ± 41.2 (55.9)	481.5 ± 28.0 (55.7)	447.8 ± 4.7 (54.9)	480.1 ± 3.7 (55.9)
***γ*-linolenic acid (C18:3 n-6)**	27.1 ± 0.7 (3.3)	30.0 ± 3.0 (4.0)	27.1 ± 1.6 (3.1)	34.4 ± 0.4 (4.2)	36.8 ± 0.3 (4.3)
***α*-linolenic acid (C18:3 n-3)**	136.2 ± 3.3 (16.7)	125.1 ± 12.9 (16.7)	158.6 ± 9.1 (18.4)	143.5 ± 1.5 (17.6)	155.6 ± 1.2 (18.1)
**Stearidonic acid (C18:4 n-3)**	9.6 ± 0.2 (1.2)	9.7 ± 1.0 (1.3)	9.4 ± 0.6 (1.1)	10.4 ± 0.1 (1.3)	12.8 ± 0.1 (1.5)
**Arachidic acid (C20:0)**	7.3 ± 0.2 (0.9)	7.0 ± 0.7 (0.9)	7.5 ± 0.4 (0.9)	7.9 ± 0.1 (1.0)	7.7 ± 0.1 (0.9)
***cis*-11-Eicosenoic acid (C20:1 n-9)**	3.6 ± 0.1 (0.4)	3.3 ± 0.3 (0.4)	3.0 ± 0.2 (0.5)	3.8 ± 0.1 (0.5)	3.7 ± 0.1 (0.4)
***cis*-11,14-Eicosadienoic acid (C20:2)**	0.4 ± 0.3 (0.0)	1.1 ± 0.8 (0.1)	1.6 ± 1.9 (0.2)	0.7 ± 0.0 (0.1)	0.2 ± 0.4 (0.0)
***cis*-8,11,14-Eicosatrienoic acid (C20:3 n-6)**	-	-	-	-	-
**Henicosanoic acid (C21:0)**	-	-	-	-	-
***cis*-8,11,14-Eicosatrienoic acid (C20:3 n-3)**	-	-	-	-	-
**Arachidonic acid (C20:4 n-6)**	-	-	-	-	-
**Eicosapentaenoic acid (C20:5 n-3)**	-	-	-	-	-
**Behenic acid (C22:0)**	2.7 ± 0.1 (0.3)	0.7 ± 0.1 (0.1)	3.3 ± 0.1 (0.4)	3.7 ± 0.0 (0.5)	3.1 ± 0.0 (0.4)
**Erucic acid (C22:1 n-9)**	-	1.0 ± 0.1 (0.1)	-	-	-
**Docosadienoic acid (C22:2 n-3)**	-	-	-	-	-
**Lignoceric acid (C24:0)**	1.1 ± 0.0 (0.1)	1.3 ± 0.1 (0.4)	-	1.9 ± 0.0 (0.2)	1.4 ± 0.0 (0.2)
**Docosahexaenoic acid (C22:6 n-3)**	-	-	-	-	-
**Others**	1.1 ± 0.4 (0.1)	1.3 ± 0.4 (0.5)	0.8 ± 0.0 (0.1)	1.0 ± 0.0 (0.1)	1.3 ± 0.4 (0.2)
**Total**	814.9 (100)	747.7 (100)	863.9 (100)	815.4 (100)	859.2 (100)
**∑ SFA**	79.7 (9.8)	73.1 (9.8)	86.7 (10.0)	88 (10.8)	82.0 (9.5)
**∑ MUFA**	106.0 (13.0)	89.9 (12.0)	98.2 (11.4)	90.6 (11.1)	91.7 (10.7)
**∑ n-6 PUFA**	483.0 (59.3)	447.8 (59.9)	508.6 (58.9)	482.2 (59.1)	516.9 (60.2)
**∑ n-3 PUFA**	145.8 (17.9)	134.8 (18.0)	168 (19.4)	153.9 (18.9)	168.4 (19.6)
***ω*-6:*ω*-3 ratio**	3.3	3.3	3.0	3.1	3.1

(-)—not detected.

**Table 4 molecules-27-02339-t004:** Carotenoid, cannabinoid and terpene content in oil extracted from hemp hearts (HSHE), hemp whole seeds (HSWH), hemp cake (HSCA), hemp seed hulls (HSHU) and cold-pressed hemp oil (HSCP). Results are expressed in mg/g oil; concentration of terpenes is below the limit of quantitation (LoQ).

Terpene/Sample	HSHE	HSWH	HSCA	HSHU	HSCP
Lutein	0.001	0.002	0.125	0.011	0.026
α-Carotene	-	-	0.024	0.001	0.029
β-Carotene	-	-	0.017	0.002	0.008
Cannabidiol (CBD)	-	+	+	+	+
Cannabidiolic acid (CBDA)	-	+	0.027	0.039	+
α-Pinene	+	+	+	+	+
β-Myrcene	+	-	+	-	+
β-Pinene	+	+	+	+	+
δ-3-Carene	-	+	+	-	-
Limonene	-	+	+	-	+
p-Cymene	-	+	-	-	-
Terpinolene	+	+	+	+	+
Isopulegol	-	+	-	-	-
Geraniol	-	+	+	-	+
β-Carophyllene	+	+	+	+	+
α-Humulene	+	+	+	+	+
Guaiol	-	+	+	-	-

(+)—detected, (-)—not detected.

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
