# Peer review of "Triacylglycerols and Other Lipids Profiling of Hemp By-Products"

_molecules, 2022, doi:10.3390/molecules27072339_

Round 1

Reviewer 1 Report

Please see my attached file

Author Response

We appreciate the reviewer’s positive comments and thorough review of the manuscript. We agree with the comments made by the reviewer and revised the manuscript accordingly. The following changes were made to the manuscript addressing the recommendations made by the reviewer.

  1. As suggested by the reviewer we have changed the title of the manuscript to “Lipid profiling of hemp by-products” which primarily stresses on lipid characterization of the hemp by-products derived from hemp seeds i.e., hemp cake and hemp seed hulls. We believe that the results of DPPH radical scavenging activity, a simple and widely used chemical assay, will certainly provide readers with the free radical scavenging potency of the oil extracted from hemp by-products. The assay is MeOH-based and the lipophilic nature of oil required to extract by MeOH prior to the assay. We followed a similar extraction method reported in the literature to determine DPPH radical scavenging activity of various commercial oils (Xuan, T.D.et al., Foods 2018, 7, 21). The extraction procedure in the experimental section and the MeOH extraction yields (ranges 4.4 – 11.0%) were added in the results section. We agree with the reviewer’s point that the DPPH radical potency of lipids was mainly due to minor components present within the oil but we believe adding percentage yield of MeOH extract covers the reviewer’s point. Moreover, pigments and cannabinoids analyses were also based on MeOH extraction as described in the experimental section (4.7 & 4.8). Thus having DPPH radical scavenging data in the manuscript makes it more relevant.
  2. The word “aquafeed” was removed from the title and the keywords section as suggested by the reviewer. Additional statements were added in the discussion section explaining the importance of hemp by-products oil uses in aquafeed application.
  3. Antioxidant activity was removed from the keywords section.
  4. New statement was added in the discussion section on how the individual TAGs were identified within overlapping signals in LC/MS analysis. MS chromatogram was provided in the supplementary documents as Figure S6 showing major TAGs ammonium adducts ions eluted at 2.35 min of hemp cake oil.
  5. A new reference was added in the discussion section which described the chemical shifts of all protons present in triglycerides.
  6. To avoid confusion the abbreviation used for neutral lipid is removed throughout the manuscript. NL is used only for neutral loss in Figure 5. We appreciate reviewer’s thorough review of the manuscript.
  7. The abbreviation TIC was corrected as Total ion current (TIC) chromatogram
  8. Page 7 and Table 4 (page 9). “d-Limonene” (delta-limonene) is corrected as “limonene”. The correction was also made throughout the manuscript.
  9. The Table 4 was revised by deleting the terpenes that were not detected. Ocimene (also known as 3,7-dimethyl-1,3,6-octatriene) and nerolidol (also known as peruviol or (3S,6Z)-3,7,11-Trimethyl-1,6,10-dodecatrien-3-ol) are a naturally occurring monoterpene and sesquiterpene respectively. Instead of cis and trans-ocimene we have written 1 and 2. Since these compounds were deleted from the table no more action is needed.
  10. The minor spelling mistakes were corrected i.e., Terpene - part 2.6, title; Tor tambroides – in discussion.

Minor changes were also made throughout the manuscript. UPLC/HRMS method for cannabinoids analysis was added in the experimental section. Numbering of references are updated accordingly.

Reviewer 2 Report

The reviewed manuscript is a well-thought-out analysis of the lipophilic constituents in various hemp products. These products are intended to be used in the nutrition of fish. The scientific assumptions of this study are perfectly correct. I have only two concerns:

  1. The authors did not refer to the publication in Frontiers in Veterinary Science - https://doi.org/10.3389/fvets.2020.572906
  2. 2. I did not find information on the toxicity tests of the analyzed products.

Author Response

Thank you for your comments, following changes were made on the revised manuscript.

  1. A new reference by Semwogerere et al., 2020 (doi.org/10.3389/fvets.2020.572906) was added, describing nutrient and phytochemical composition of hemp by-products, their bioavailability, and bioefficacy.
  2. Even though no toxicity studies have been reported on hemp oils, we added a statement describing antinutrient components present in hemp seeds in the discussion section with additional references.

Round 2

Reviewer 1 Report

Some minor changes should be done in the text.

  1. My suggestion for Title is: "Triacylglycerols and other lipids profiling of hemp by-products".
  2. Page 7, line 177: "δ-Limonene" (delta-Limonene).

  3. "TAGs" is an abbreviation of “TriAcylGlycerols”, not triglycerides.

  4. In Abstract, full name of "TAGs" is needed for understanding.

Author Response

Thank you for your comments, following changes were made on the revised manuscript.

  1. Title of the manuscript is changed to “Triacylglycerols and other lipids profiling of hemp by-products”.
  2. d-Limonene is changed to delta-Limonene in page 7.
  3. Throughout the manuscript word triglycerides are changed to triacylglycerols (TAGs)
  4. In the abstract full name of TAGs i.e., triacylglycerols (TAGs) is added.